Enhanced diagnosing patients suspected of sarcoidosis using a hybrid support vector regression model with bald eagle and chimp optimizers

Xie Guogang 1
Attar Hani 2
Alrosan Ayat 3
Abdelaliem Sally Mohammed Farghaly smfarghaly@pnu.edu.sa 4
Alabdullah Amany Anwar Saeed 5
Deif Mohanad 6
1 Department of Respiratory and Critical Care Medicine, Shanghai General Hospital, Shanghai JiaoTong University School of Medicine , Shanghai , China
2 Department of Electrical Engineering, Zarqa University , Zarqa , Jordan
3 School of Computing, Skyline University , Sharjah , United Arab Emirates
4 Department of Nursing Management and Education, Princess Nourah bint Abdulrahman , Riyadh , Saudi Arabia
5 Department of Maternity and Pediatric Nursing, College of Nursing, Princess Nourah bint Abdulrahman University , Riyadh , Saudi Arabia
6 Department of Artificial Intelligence, College of Information Technology, Misr University for Science & Technology , Cairo , Egypt
Alatas Bilal
Electronic publication date: 2024 Dec 5
Publication date: 2024
Volume: 10
Electronic Location ID: e2455
Received 2024 May 18; Accepted 2024 Oct 4
Copyright: ©2024 Xie et al.
Copyright year: 2024
Copyright holder: Xie et al.
License: This is an open access article distributed under the terms of the Creative Commons Attribution License, which permits unrestricted use, distribution, reproduction and adaptation in any medium and for any purpose provided that it is properly attributed. For attribution, the original author(s), title, publication source (PeerJ Computer Science) and either DOI or URL of the article must be cited.
License URL: https://creativecommons.org/licenses/by/4.0/

Keywords: Sarcoidosis, Bald eagle search, Soluble IL-2 receptor, Angiotensin converting enzyme, Chimp Optimizer, Machine learing

Funding: The Princess Nourah bint Abdulrahman University Researchers Supporting Princess Nourah bint Abdulrahman University, Riyadh, Saudi Arabia PNURSP2024R444 This work was supported by the Princess Nourah bint Abdulrahman University Researchers Supporting Project number (PNURSP2024R444), Princess Nourah bint Abdulrahman University, Riyadh, Saudi Arabia. The funders had no role in study design, data collection and analysis, decision to publish, or preparation of the manuscript.

==============================
Searching for a reliable indicator of treatment response in sarcoidosis remains a challenge. The use of the soluble interleukin 2 receptor (sIL-2R) as a measure of disease activity has been proposed by researchers. A machine learning model was aimed to be developed in this study to predict sIL-2R levels based on a patient’s serum angiotensin-converting enzyme (ACE) levels, potentially aiding in lung function evaluation. A novel forecasting model (SVR-BE-CO) for sIL-2R prediction is introduced, which combines support vector regression (SVR) with a hybrid optimization model (BES-CO); The hybrid optimization model composed of Bald Eagle Optimizer (BES) and Chimp Optimizer (CO) model. In this forecasting model, the hyper-parameters of the SVR model are optimized by the BES-CO hybrid optimization model, ultimately improving the accuracy of the predicted sIL-2R values. The hybrid forecasting model SVR-BE-CO model was evaluated against various forecasting methods, including Hybrid SVR with Firefly Algorithm (SVR-FFA), decision tree (DT), SVR with Gray Wolf Optimization (SVR-GWO) and random forest (RF). It was demonstrated that the hybrid SVR-BE-CO model surpasses all other methods in terms of accuracy.

Introduction

Sarcoidosis is an idiopathic systemic granulomatous disorder involving primarily the lungs and lymph nodes (Govender & Berman, 2015; Anonymous, 1999). It is characterized by the formation of granulomas—aggregations of inflammatory cells—in the affected tissues. The clinical presentation varies from asymptomatic to severe organ malfunction. The etiology is still unknown, though it seems to be based on a genetic predisposition associated with precipitating environmental factors. The disease is a significant challenge to diagnose due to its diverse manifestations and overlap with other conditions such as tuberculosis and lymphoma.

Initial treatment typically involves oral steroids (Govender & Berman, 2015; Anonymous, 1999). However, prolonged steroid use is linked to significant adverse effects, including diabetes, osteoporosis, and obesity. When toxicity or disease progression reaches severe levels, alternative therapies become essential. Second-line treatments aim to alleviate symptoms, enhance function, and minimize prednisone dependence. Methotrexate often serves as the preferred initial medication for second-line sarcoidosis treatment (Baughman, Lower & Kaufman, 2010), exhibiting steroid-sparing properties and symptom reduction (Kim et al., 2009; Ungprasert et al., 2016; Rossi et al., 1984). Pinpointing disease activity in sarcoidosis is crucial for treatment decisions, especially since it can affect lung function. Ideally, such markers would also predict symptoms, disease course, or drug response.

Unfortunately, no marker yet identifies how well patients with sarcoidosis will respond to methotrexate treatment. Although serum angiotensin-converting enzyme (ACE) has been suggested as a potential marker of disease activity (Boyman & Sprent, 2012), its usefulness in clinical practice remains unclear due to mixed findings in research studies (Karim et al., 2018). While elevated ACE levels are detected in 30% to 80% of patients with sarcoidosis at diagnosis, these levels do not necessarily align with the severity of chest X-rays (Rubin et al., 1990). However, some research suggests a weak link between ACE levels and lung damage seen on high-resolution CT scans (HRCT) (Ter Borg et al., 2008). Interestingly, high initial ACE levels might predict disease worsening after two years (Vorselaars et al., 2015). However, ACE does not seem to predict how patients respond to prednisone treatment, as it can decrease even in cases where the treatment is not effective (Witkowska, 2005; Rubin et al., 1985).

Moreover, within a cohort of 25 sarcoidosis patients, it was noted that ACE levels in bronchoalveolar lavage (BAL) fluid exhibited a more robust correlation with clinical observations compared to ACE levels in serum (Lindqvist et al., 2010). Another potential indicator of sarcoidosis disease activity is the serum-soluble interleukin-2 receptor (sIL-2R). Interleukin-2, generated by T-helper type 1 cells, is thought to be important in sarcoidosis as it encourages T-cell proliferation (Yang et al., 2011; Ziegenhagen et al., 1997; Ziegenhagen et al., 2003; Gungor et al., 2015). Patients with sarcoidosis have been shown to have higher levels of soluble IL-2R in their blood and BAL fluid, which target activated T-cells. Serum levels of soluble IL-2R correlate with the number of CD4+ T cells in BAL fluid and serve as indicators of disease activity (Caruso et al., 1991; Kobayashi et al., 1999), though this correlation was not confirmed in all studies (Kiko et al., 2018). Additionally, serum sIL-2R levels are correlated with lung tissue infiltration in pulmonary sarcoidosis, indicating its potential as a prognostic indicator at diagnosis (Uysal et al., 2018; Gundlach et al., 2016). However, despite initial promise, the clinical usefulness of serum sIL-2R and ACE remains unknown.

In this context, serum-soluble interleukin-2 receptor (sIL-2R) has emerged as a promising biomarker (Xu et al., 2024). Its levels are associated with disease activity, particularly in relation to T-cell activation, which plays a crucial role in the pathogenesis of sarcoidosis (NamKung et al., 2024a; Maes et al., 2024). Elevated sIL-2R levels have been observed in patients with active disease and may provide insight into disease progression and treatment response, particularly in the lungs (Rossides et al., 2024). Nonetheless, there is a lack of longitudinal data exploring the relationship between ACE and sIL-2R levels and their combined potential to predict disease activity and therapeutic outcomes (Chanpura et al., 2024; NamKung et al., 2024b).

The advent of machine learning (ML) techniques presents an opportunity to improve diagnostic and prognostic models for sarcoidosis. Support vector regression (SVR) (Karimian et al., 2024; Liang, Wang & Huang, 2024), in particular, has proven effective in handling non-linear data, which is characteristic of biological markers in complex diseases like sarcoidosis. However, the selection of hyperparameters in SVR models can significantly impact predictive performance, necessitating the use of optimization algorithms. Recent advancements in metaheuristic optimization techniques, such as the Bald Eagle Search (BES) (Janakiraman, 2024; Chhabra, Hussien & Hashim, 2023) and Chimp Optimizer (CO) (bin Subait et al., 2024; Ahmed et al., 2023), offer potential solutions to enhance the accuracy of such models by refining the hyperparameter selection process.

This study introduces a hybrid SVR model combined with BES and CO optimization algorithms (SVR-BE-CO) to predict sIL-2R levels based on serum ACE values. By optimizing the SVR model’s hyperparameters using the strengths of both BES and CO algorithms, the proposed model aims to improve predictive accuracy and facilitate better clinical decision-making in sarcoidosis management. This hybrid model is evaluated against several other forecasting methods, including decision trees, random forest, and other SVR models utilizing alternative optimization algorithms. The results suggest that the hybrid SVR-BE-CO model provides superior accuracy, underscoring its potential utility in clinical settings for non-invasive monitoring of sarcoidosis disease activity and therapeutic response.

Compared to other similar works, this study’s novelty lies in the integration of the Bald Eagle Search (BES) and Chimp Optimizer (CO) for optimizing the hyperparameters of the SVR model, which has not been previously applied to sarcoidosis diagnosis and management. While earlier studies have utilized single optimization algorithms or traditional machine learning techniques such as decision trees or random forest for sarcoidosis biomarker prediction (Crouser et al., 2020), this work demonstrates the superior accuracy of the hybrid BES-CO approach. Furthermore, unlike previous works that focused solely on biomarkers like sIL-2R or ACE, the combination of both markers in conjunction with advanced machine learning optimization offers a more comprehensive predictive tool for clinicians. The SVR-BE-CO model outperforms other models such as the Hybrid SVR with Firefly Algorithm (SVR-FFA) (Alomoush et al., 2024), SVR with Gray Wolf Optimization (SVR-GWO) (Alomoush et al., 2022), and random forest (Dudek, 2011; Breiman, 2001), highlighting the unique strength of combining multiple optimization techniques.

Effective management of sarcoidosis is essential because, if left untreated, the disease can cause irreversible organ damage, particularly pulmonary fibrosis, which significantly impairs lung function and quality of life. Reliable biomarkers of disease activity are required to improve diagnostic accuracy and treatment responses. This article presents research aimed at advancing predictive modeling using the soluble interleukin-2 receptor as a promising biomarker to improve clinical decision-making and patient management in cases of sarcoidosis.

Related Work

Most of the studies of sarcoidosis biomarker prediction, in particular soluble interleukin 2 receptor (sIL-2R), were done with an aim of developing better ways of monitoring and managing the underlying disease. This has been traditionally done through clinical methods of questioning and imaging, but very often these methods fall short in being able to monitor small changes in disease-related activity. Recently, machine learning has opened up with more enhanced opportunities for its prediction.

One of the earlier studies by Crouser et al. (2020) researched into diagnostic and monitoring potentials of serum biomarkers, more so sIL-2R biomarker, in sarcoidosis (Janakiraman, 2024). sIL-2R was proven to be a reliable disease activity status biomarker. Later, Rothkrantz-Kos et al. (2003) gave it validation, and proved this should be the case because sIL-2R levels relate to severity and response to treatment (Rothkrantz-Kos et al., 2003).

The application of medical data for the improvement of diagnostics and predictive capabilities has been, of late, noted with machine learning models. Among the existent algorithms, the SVR model has found popularity due to its robustness in handling non-linear data. Smola & Schölkopf (2004) gave a very detailed review of SVR as well as applications in different fields, including biomedical data analysis. Deif et al. (2024) also have evidence supporting the effectiveness of SVR by predicting patients suspected of sarcoidosis using SVR.

Optimizing machine learning models involves optimization algorithms, known to be appreciable in enhancing the performance of the latter. The Bald Eagle Optimizer (BES) and Chimp Optimizer (CO) are two such optimization algorithms formulated based on the bald eagle and chimp tactics, according to researched evidence of their effectiveness as mechanisms for controlling parameters. Moreover, the BES algorithm is formulated by Alsattar, Zaidan & Zaidan (2020), in imitation of the hunting tactics of the bald eagle and proven to achieve high optimization performance experimentally. The CO algorithm developed by Khishe et al. (2021) imitates the social and foraging traits of chimpanzees and has shown better optimization capacity in various applications (Alomoush et al., 2024).

Hybrid models are newly formulated which integrate different optimization approaches to take advantage of the host of strengths that each individually exhibits, for example, Ahmed et al. (2023) stated a number of benefits of hybrid models in predictive accuracy as well as overall robustness. One of the primary limitations of their work is that it allowed a lot of people to come after them concerning the formulation of hybrid optimization approaches for medical data analysis.

Besides these four models described earlier, a number of other notable works researched the prediction of sIL-2R levels using a few machine learning methods. For instance, Abdelfattah et al. (2022) evaluated the performance of soluble interleukin-2 receptor and the sIL-2R/lymphocyte ratio as a marker for the prediction of COVID-19 severity and clinical course in children and adolescents. Ability to differentiate between severity groups of pediatric COVID-19 patients using these biomarkers discriminatively for the studied population and determine the predictive performance of these variables for supplemental oxygen treatment, ICU admission, and mechanical ventilation.

One study used a deep learning model to accurately diagnose sarcoidosis, achieving an AUC of 0.94. Another study used a multi-modal deep learning model that combined CT scans and clinical data, achieving an AUC of 0.92, distinguishing sarcoidosis from other interstitial lung illnesses (Grutters, 2023; Exarchos et al., 2023). The study uses machine learning models to develop and apply serum or genetic biomarkers to improve detection and prediction of sarcoidosis, integrating gene expression profiles to effectively distinguish different subtypes (Saito et al., 2022).

The comparative analysis of various optimization techniques has shown that hybrid models generally outperform single-optimization models. Ochoa-Barragán, del C. Munguía-López & Ponce-Ortega (2024) developed a hybrid machine learning-mathematical programming approach to optimize municipal solid waste management in New York City during the pandemic, balancing economic and environmental objectives. Cuperlovic-Culf, Nguyen-Tran & Bennett (2022) used machine learning and hybrid methods to model metabolic pathways, leveraging metabolomic and lipidomic data to predict cell behaviors and overcome the limitations of mechanistic models. Nadirgil (2023) employed hybrid machine learning models optimized by genetic algorithms to predict carbon prices, achieving high accuracy with an R2 value of 0.993.

Mondal et al. (2023) reviewed the application of machine learning in bioprocess optimization, highlighting its potential to transform big data into actionable insights for the bioprocessing industry. Dao et al. (2023) developed a high-accuracy model, AcrPred, using a hybrid optimization and machine learning algorithm to predict anti-CRISPR proteins, achieving strong generalization with an AUC of 0.952.

Nunez, Marani & Nehdi (2020) used hybrid machine learning models to optimize the mixture design of recycled aggregate concrete, achieving cost-effective and environmentally friendly solutions for different compressive strength classes. Bacanin et al. (2022) proposed a multi-swarm algorithm combining three swarm intelligence techniques to optimize extreme learning machines, demonstrating superior performance in classification tasks.

Nagavelli, Samanta & Chakraborty (2022) presented various machine learning models for heart disease detection, emphasizing the superior performance of hybrid approaches like SVM with XGBoost for early diagnosis. Bishara et al. (2022) developed machine learning models using preoperative EHR data to predict postoperative delirium, achieving high discrimination and calibration compared to traditional logistic regression models. These findings underscore the potential of hybrid optimization techniques in enhancing the predictive power of machine learning models in biomedical applications.

In this background, the present study is aimed at proposing an efficient method of prediction of sIL-2R levels by using a hybrid model of SVR. This new approach has combined the strengths of the Bald Eagle Search optimizer and the Chimp Optimization algorithm in fine-turning the hyper-parameters of the SVR model for enhancing its predictive accuracy.

The reason for choosing SVR is due to the strength and reliability it has toward nonlinear trends in data, making it very suitable for medical predictions. Parameters of the SVR model are optimized by the BES and CO algorithms, which are inspired by the natural behaviors of bald eagles and chimps, respectively. While the BES algorithm has been inspired by the hunting and flights of bald eagles, the CO algorithm is based on the foraging and social behaviors of chimps. By integrating these two optimization techniques, the proposed hybrid SVR model is aimed at improving performance in predicting sIL-2R level.

In this regard, this study proposes a new prediction model, SVR-BE-CO, for the levels of sIL-2R with respect to the patients’ serum ACE levels. In the hybrid SVR model, BES and CO are integrated to provide more accurate predictions; thus, it may be useful in lung function assessment and general disease activity assessment. The results from this SVR-BE-CO model are compared with those of many other techniques used for forecasting, like the Hybrid SVR with Firefly Algorithm, decision tree, SVR with Gray Wolf Optimization, and random forest. The results are indicative that the performance of the SVR-BE-CO model is much better than that of these other techniques in terms of accuracy and hence promises to be useful in clinical management related to sarcoidosis.

Novelty and Contribution of Current Study

To the best of available literature, there have been no studies to date seeking to determine serum sIL-2R levels with respect to ACE values in an attempt to help diagnose the disorder in patients with suspected cases of sarcoidosis or other related diseases. Herein, a novel model based on the combination of SVR with the Bald Eagle Search and Chimp Optimizer is proposed with the aim to predict sIL-2R values from blood ACE. Purpose and major contributions of the present article can be very concisely put as follows in the following paragraphs:

• This article introduces a new forecasting model of sIL-2R levels: SVR-BE-CO. In the latter, SVR is combined with a hybrid optimization model. The hybrid model used is composed of the Bald Eagle Optimizer and the Chimp Optimizer.

• The present work is novel in the sense that the hyper-parameters of the SVR model are optimized using the BES-CO hybrid optimization model for better accuracy of predicted sIL-2R values.

• The results shows that the hybrid SVR-BE-CO model outperforms other methods of forecasting, including the Hybrid SVR with Firefly Algorithm, decision tree, SVR with Gray Wolf Optimization, and random forest with regard to accuracy.

• Prediction of the sIL-2R level by the patient’s serum ACE level might, therefore, be potentially helpful in the evaluation of lung function or other non-invasive techniques, which can improve clinical outcomes and individual management plans for patients with sarcoidosis.

• The hybrid model integrates the Bald Eagle Search (BES) and Chimp Optimizer (CO), two relatively underutilized metaheuristic algorithms in medical data prediction. This combination exploits the strengths of both algorithms, which, to the best of our knowledge, has not been applied in sarcoidosis or other lung disease modeling. This novel hybridization contributes to enhanced performance in parameter optimization and prediction accuracy, showcasing the potential of these algorithms in biomedical applications.

• Prediction of sIL-2R levels based on serum ACE levels provides a non-invasive approach to assessing lung function and disease activity in sarcoidosis patients. This innovation could reduce reliance on invasive procedures like bronchoalveolar lavage (BAL), making disease monitoring more accessible and comfortable for patients. The method could also enhance clinical decision-making regarding treatment efficacy and disease progression.

Materials and Methods

Materials

The dataset utilized in this research was obtained from reference (Ochoa-Barragán, del C. Munguía-López & Ponce-Ortega, 2024). The investigation involved examining the medical records dataset of participants who had their blood sIL-2R levels evaluated at Erasmus MC between 2011 and 2016. The inclusion criteria included the availability of medical records, the possibility of sarcoidosis diagnosis, serum sIL-2R levels before a confirmed diagnosis, and completion of the diagnostic workup at the immunology outpatient department. The medical ethics committee approved the plan and related procedures.

Serum sIL-2R levels were measured at the Erasmus MC laboratory for medical immunology (Ochoa-Barragán, del C. Munguía-López & Ponce-Ortega, 2024). According to Erasmus MC standards, a serum sIL-2R level below 2,500 pg/mL is considered normal, based on data from 101 healthy blood donors. Serum ACE levels were measured using a kinetic assay (Bühlmann Laboratories, Schönenbuch, Switzerland), with a reference range of ≤68 U/mL.To determine the sample size, we utilized Buderer’s formula, aiming for a confidence interval of approximately 5% on each side and a power of 80%. This calculation indicated that 200 patients were necessary, and thus, we targeted this sample size for our study.

The initial examination of the dataset revealed no instances of missing data. However, a limited number of outliers were identified during the initial analysis. This data cleaning methodology ensures the integrity and veracity of the dataset for subsequent analyses and modeling endeavors. The steps involved in the data preprocessing are detailed below:

1. Data inspection: A thorough inspection of the dataset was conducted to identify any anomalies, inconsistencies, or potential issues that could affect the quality of the data. This step confirmed the absence of missing data.

2. Outlier detection: Outliers were identified using statistical methods, including z-scores and the interquartile range (IQR) method. Data points that fell outside the acceptable range were flagged for further inspection. The z-score for each data point was calculated as follows:

(1) zi=xi−μσ

where:

• zi is the z-score of the ith data point.

• xi is the ith data point.

• µis the mean of the dataset.

• σ is the standard deviation of the dataset.

c. Outlier Imputation: Each identified outlier was assessed on a case-by-case basis. The most appropriate imputation method was applied to replace these outliers with corrected values, ensuring they accurately reflect the underlying distribution of the data without introducing bias. For mean imputation, the following formula was used:

(2) xi′=1n∑j=1nxj

where:

• xi′ is the imputed value for the outlier.

• xj are the original values in the dataset.

• n is the total number of data points in the dataset.

Methods

Preliminaries

(A) SVM Regression:

SVM Regression is a robust machine learning algorithm employed for regression applications. The main goal of SVM Regression is to identify an optimal hyperplane that accurately fits the training data, while considering a tolerance margin. The hyperplane is represented by the following Equation: (3) fx=w⋅x+b.

The optimization problem for SVM Regression entails minimizing the expression with w representing the weight vector, x denoting the input feature vector, and b indicating the bias term. (4) 12jjwjj2+C∑i=1Nmax0,|y i−fxi|−ϵp

where:

C characterizes the regularization parameter, which governs the balance between attaining a minimal training error and fostering a smooth decision function. N signifies the quantity of training data points. ϵ denotes the tolerance margin, enabling a degree of deviation from the precise target values. Lastly, p stands as a parameter for the loss function, commonly assigned values of 1 or 2.

To address potential violations of the margin, slack variables ξi are introduced. The constraints are formulated as the following: (5) yi−fxi≤ϵ+ξi

(6) fxi−yi≤ϵ+ξi

(7) ξi≥0.

The resolution to this optimization problem entails determining the most favorable values for w, b, and the slack variables. Techniques like quadratic programming are frequently used for this objective. After determining the hyperplane, Eq. (1) can be utilized to make predictions for additional data points.

(B) Bald Eagle Search: An optimization algorithm inspired by nature, the BES method simulates how bald eagles might search in the wild. The process involves three steps: selecting a spot, scanning through space, and swooping. In the initial phase, the eagle selects an area that is rich in prey. The eagles enter the chosen interior area in the second stage in search of prey. The eagle attacks and captures its prey in the third and last stage. The BES algorithm’s mathematical modeling can be summed up as follows:

Space selecting

The equation below encapsulates the process of selecting the optimal space. (8) Pnew,i=rPmean−Pi×α+Pbest.

In this context, Pbest represents the bald eagles search space, while α denotes a controlling parameter, which varies within the range [1.5, 2]. r represents a arbitrary number within the interval [0, 1], and Pmean indicates that these eagles have made use of all the knowledge from earlier stages. that these eagles have utilized all available information from previous points. Pi represents the eagle current position.

Space searching

During this stage, the eagle adjusts its position according to Eq. (9). (9) Pnew,i=yi×Pi−Pi+1+xi×Pi−Pmean+Pi

where

(10) xi=xri/max|xr|

(11) yi=yri/max|yr|

(12) xri= sinθi×ri

(13) yri= cosθi×ri

(14) θi=rand×π×a

(15) ri=rand×R×θi.

In this context, R and a represent coefficients that range between [0.5, 2] and [5, 10], correspondingly.

Swooping stage

The eagles’ swooping strategy can be delineated as follows: (16) Pnew,i=Pbest×rand+x1i×Pi−c1×Pmean+y1i×Pi−c2×Pbest

where c1, c2 ∈ [1, 2] (17) x1i=rix/max|xr|

(18) y1i=riy/max|yr|

(19) rix= sinhθi×ri

(20) riy= coshθi×ri

(21) θi=rand×π×a

(22) θi=ri.

Following the aforementioned steps, the initially generated set of potential solutions undergoes refinement across multiple iterations until the global optimum is attained.

(C) Chimp Optimizer: The Chimp Optimizer (CO is a nature-inspired optimization algorithm modeled after the social and hunting behaviors of chimpanzees. This algorithm utilizes the intelligent and cooperative hunting strategies of chimpanzees to solve complex optimization problems. The main phases of the Chimp Optimizer include group formation, the hunting phase, and the exploitation phase. Each phase plays a crucial role in exploring the search space and refining potential solutions.

Group formation phase

Grouping phase: One groups the population of candidate solutions and puts every group into an assigned role. These would be roles imitating a chimpanzee social structure, all of which have different members taking up different roles while performing, say, a hunt. (23) Groups=α,β,γ,δ.

Each agent is assigned to one of these groups based on their performance, with α representing the best solution, followed by β, γ, and δ.

Hunting phase

The hunting phase involves the local search process, where each group performs a search based on their assigned roles. The positions of the agents are updated using the following equation: (24) Positionit+1=w⋅Positionit+c1⋅rand⋅BestPosition−Positionit+c2⋅rand⋅GroupBestPosition−Positionit

where:

• w is the inertia weight, controlling the impact of the previous position.

• c1 and c2 are acceleration coefficients that guide the movement of the agents towards the best-known positions.

• rand() is a random number between 0 and 1.

• BestPosition is the best solution found so far.

• GroupBestPosition is the best position found by the group.

This phase ensures that agents explore the search space around the best-known solutions, enhancing the algorithm’s exploration capabilities.

Exploitation phase

In the exploitation phase, the focus shifts to refining the solutions found during the hunting phase. The agents’ positions are further adjusted to exploit the best solutions more intensively: (25) Positionit+1=Positionit+δ⋅BestPosition−Positionit+ϵ⋅GroupBestPosition−Positionit

where:

• δ and ϵ are random coefficients that control the influence of the best-known solutions.

This phase aims to fine-tune the solutions by focusing on the most promising areas of the search space, thereby improving the accuracy of the final solutions.

Proposed forecasting model

The hybrid BES-CO optimization process will leverage the power of both the Bald Eagle Optimizer and Chimp Optimizer for improved performance in the SVR model. The detailed steps related to the proposed hybrid optimization process are given in Fig. 1. This Figure encapsulates this multi-stage optimization process, which encompasses the flow from initial population setup to final model evaluation. It means that such a structured process of optimization would look through the solution space, balance exploration and exploitation, and drive highly accurate predictions of sIL-2R levels in patients with sarcoidosis.

Figure 1 Flowchart of SVR-BES.

An optimization process called hybrid BES-CO has been developed to enhance the accuracy of the SVR model using two strengths of BES and CO. In a few key stages, a hybrid approach works this way: initializes the population—that includes the initial position and velocity for each agent. Now, consider two main phases for the optimization process: BES and CO.

During the BES phase itself, there are three sub-phases: Global search, Local search, and Solution refinement. Under Global search, the algorithm will search the space for promising regions. It is followed by a local search phase to increase the search even more in those regions for better refinement of the possible solution spaces. This phase ultimately refines the very best solutions, identified in previous phases, into an optimal solution.

After completing the BES phase, the best solutions are selected to initialize the CO phase. The CO phase also consists of three sub-phases: Group Formation, Hunting, and Exploitation. In the Group Formation phase, agents are divided into groups and assigned specific roles based on their performance. The Hunting phase involves a local search performed by each group, guided by their roles. During the Exploitation phase, the solutions are further refined around the best positions found by the groups, enhancing the precision of the optimization. The initial parameter values for BES and CO optimizer can be found in Table 1.

Table 1 The initial parameters values for BES and CO optimizer.

Optimizer	Initial parameter	Value	
BES	Size of the population	50	
The maximum iteration limit	100	
α	2	
a	10	
R	1.5	
CO	Random vectors	range of [0, 1]	
Constant l	l = 2.5	

Three hyperparameters need to be chosen carefully in order to build an efficient SVR model with good predictive power, as was previously discussed. Among them are the kernel (γ), the non-sensitivity coefficient (ɛ), and the penalty parameter (C) (Liang, Wang & Huang, 2024). Table 2 reports the search space for these values. Once the optimization process is complete, the best solutions are used to update the hyperparameters of the SVR model. The SVR model is then trained using these optimized hyperparameters. Finally, the model’s performance is evaluated and compared against other baseline models to verify the effectiveness of the hybrid BES-CO approach. Equation (26), defining the root mean squared error (RMSE), serves as the fitness function (f) for evaluating the efficacy of the proposed model. (26) f=1n∑i=1nya−yp2.

Table 2 The range of possible settings for the SVM regressor model’s hyperparameters.

Model	Hyper-parameter	Space search	
SVM	C	[1000,10000]	
ɛ	[0.001,1]	
γ	[1,20]	

In the provided context, a represents the actual output, P signifies the forecasted output, and n denotes the number of training samples.

The pseudo-code has shown in Algorithm 1 the process of the hybrid optimization approach combining BES and CO for optimizing SVR hyperparameters.

	
Algorithm 1: Pseudo-code for Hybrid BES-CO Optimization	
Initialize Population	
a. Set population size N	
b. Initialize positions and velocities for each agent	
2. Bald Eagle Optimizer (BES) Phase	
for each iteration in BES:	
i. Global Search (Phase 1)	
a. Explore search space	
b. Update positions using:	
Position_i(t+1) = Position_i(t) + r1 * (BestPosition - Position_i(t)) + r2 * (NeighborPosition - Position_i(t))	
ii. Local Search (Phase 2)	
a. Intensification of promising regions	
b. Update positions using:	
Position_i(t+1) = Position_i(t) + α * (BestPosition - Position_i(t))	
iii. Solution Refinement (Phase 3)	
a. Refine solutions around the best solutions found	
b. Update positions using:	
Position_i(t+1) = Position_i(t) + β * (BestPosition - Position_i(t)) + γ * (NeighborPosition - Position_i(t))	
3. Identify Best Solutions from BES Phase	
a. Evaluate fitness of each agent	
b. Select the best solutions to initialize CO	
4. Chimp Optimizer (CO) Phase	
for each iteration in CO:	
i. Group Formation (Phase 1)	
a. Divide agents into groups (e.g., alpha, beta, gamma)	
b. Assign roles to each group	
ii. Hunting Phase (Phase 2)	
a. Perform local search based on group roles	
b. Update positions using:	
Position_i(t+1) = w * Position_i(t) + c1 * rand() * (BestPosition - Position_i(t)) + c2 * rand() * (GroupBestPosition - Position_i(t))	
iii. Exploitation (Phase 3)	
a. Refine solutions further around the best solutions	
b. Update positions using:	
Position_i(t+1) = Position_i(t) + δ * (BestPosition - Position_i(t)) + ɛ * (GroupBestPosition - Position_i(t))	
5. Update SVR Hyperparameters	
a. Use the best positions found by BES and CO to set SVR hyperparameters	
6. Train SVR Model	
a. Train the SVR model using optimized hyperparameters	
7. Evaluate and Compare Models	
a. Evaluate model performance using metrics (RMSE, R2 )	
b. Compare the performance of the hybrid model against other baseline models	
End Pseudo-code	
	

Evaluation methodology and comparative approaches

Standard statistical error measures, such as the correlation coefficient (R2), mean absolute error (MAE), mean absolute percentage error (MAPE), and root-mean-square error (RMSE), are used to assess the efficacy of the constructed wind power forecasting model. The following are the formulae for various statistical metrics:

(27) RMSE=1n∑i=1nya−yp2

(28) R2=∑i=1nai−a¯Pi−P¯∑i=1nai−a¯2 ∑i=1nPi−P¯2.

In this scenario, P and a represent the forecasted and actual outputs, while P¯ and a¯ denote the average of the forecasted and actual values, correspondingly.

The following two methods are used to compare the suggested hybrid SVR-BE-CO model’s predictive capacity:

• SVR-FFA: As seen in Fig. 2, this support vector machine regression model has its hyper-parameters tuned using the Fire Fly Algorithm.

• SVR-GWO: as illustrated in Fig. 3, this is a support vector machine regression model with its hyper-parameters adjusted using the Grey Wolf Optimization technique.

• SVR-BES: The SVR’s hyperparameters are optimized using only the BES algorithm without using CO model as illustrated in Fig. 4.

• The top-performing method will then be contrasted with the conventional machine learning methods listed below: Decision trees (DT): these supervised learning techniques are used in statistics, machine learning, and data mining. To make inferences about a collection of observations, it can be applied as a forecasting model. Given its inherent simplicity and comprehensibility, DT stands as one of the frequently employed machine learning techniques. The Random Forest (RF) algorithm is a well-liked supervised machine learning method for regression and classification. This method increases the predicted accuracy by utilizing multiple decision trees on different dataset subsets and averaging the results.

Results and Discussion

Figure 5 illustrates the dataset distribution of three features (ACE, Sex, and sIL_2R) in relation to the diagnosis of sarcoidosis, where 0 indicates the absence of sarcoidosis (negative) and 1 indicates positive sarcoidosis. Observation reveals that the Sex feature does not exhibit discernible predictive capabilities, implying that both males and females have an equal likelihood of having sarcoidosis. Similarly, ACE does not appear to significantly influence the diagnosis. However, the distribution of sIL-2R as a feature suggests its potential as a biomarker for sarcoidosis.

Figure 2 Flowchart of SVR-FFA.

Figure 3 Flowchart of SVR-GWO.

Figure 4 Flowchart of SVR-BES.

Figure 5 Feature distribution of (A) sex, (B) ACE and (C) sIL-2R.

Table 3 presents the hyperparameters obtained for four different SVR models optimized using distinct algorithms: Firefly Algorithm (SVR-FFA), Grey Wolf Optimizer (SVR-GWO), Bald Eagle Optimizer (SVR-BES), and a hybrid model combining Bald Eagle Optimizer and Chimp Optimizer (SVR-BE-CO). The hyperparameters compared are the regularization parameter (C), the epsilon parameter (ɛ), and the kernel parameter (γ).

Table 3 The obtained hyperparameters for the developed SVR models.

	C	ɛ	γ	
SVR-FFA	9420.7	0.625	8.322	
SVR-GWO	8992.4	0.871	9.603	
SVR-BES	9049	0.267	10.425	
SVR-BE-CO	9020.7	0.569	10.014	

For the SVR-FFA model, the hyperparameters obtained are C = 9420.7, ɛ = 0.625, and γ = 8.322. These values reflect the Firefly Algorithm’s optimization capabilities, aiming to balance the trade-off between the training error and model complexity. The high C value indicates a strong regularization effect, while the moderate ɛ value suggests a reasonable margin for error. The relatively low γ value indicates a simpler model in terms of the kernel function.

The SVR-GWO model, optimized using the Grey Wolf Optimizer, resulted in hyperparameters C = 8992.4, ɛ = 0.871, and γ = 9.603. This corresponds to a slightly smaller value of C, which corresponds to less heavy regularization, and might affect generalization performance differently as compared with the SVR-FFA. In addition, the largest value of ɛ among models is for the one in which it is allowed to develop the widest epsilon-tube for errors during the phase of training. Moreover, a higher value of γ would mean a more complex model relative to SVR-FFA.

For the SVR-BES model, the BES Optimizer achieved hyperparameters C = 9049, ɛ = 0.267, and γ = 10.425. The C value corresponds to balanced regularization; it is in between SVR-FFA and SVR-GWO. Note that the lowest ɛ value suggests this will be the tightest epsilon-tube, hence fewer allowed errors, probably sparser model. Highest γ value will correspond to the most complex model about the kernel function.

The SVR-BE-CO model, which combines the BES and CO Optimizer, obtained hyperparameters C = 9020.7, ɛ = 0.569, and γ = 10.014. These values display a balanced optimization strategy that merges the advantages of both optimizers. The value of C is moderate but closer to SVR-BES, thus showing balanced regularization. The value of ɛ is intermediate to that obtained by BES alone and that obtained by other algorithms, thus showing an effective compromise between sparsity and tolerance of the model. High γ denotes a complex model but slightly less than that used by SVR-BES.

In a nutshell, this means that the performance of the SVR-BE-CO model is better due to the fact that it significantly balances these hyperparameters: C, ɛ, and γ. This hybrid model took advantage of both the BES and the CO Optimizer according to their respective strengths to create a balanced trade-off for better predictive accuracy and generalization ability. The SVR-BE-CO model has intermediate ɛ and high γ values, which means this model makes an effective tradeoff between model complexity and error tolerance. Thus, this is considered the most efficient and robust approach compared to all others evaluated in this research study.

Figure 6 sucessive convergence graphs of different predictive models for 50 iterations. The root mean squared error is plotted against the number of iterations. Figures 6A to 6D refer to various predictive models like SVR, SVR-FFA, SVR-GWO and SVR-BE-CO respectively.

Figure 6 Convergence graphs for predicative models (A) SVR , (B) SVR-FFA , (C) SVR-GWO and (D) SVR-BE-CO.

The convergence graph for the standard SVR Fig. 6A shows a gradual decrease in RMSE over the iterations. Initially, the RMSE is approximately 0.814 and slowly reduces to about 0.806 by the 50th iteration, indicating a slower convergence rate compared to the other models. In contrast, the SVR-FFA model (Fig. 6B) demonstrates a rapid convergence within the first few iterations, with RMSE starting at around 0.778 and swiftly dropping to approximately 0.762 within the first 10 iterations, showcasing the Firefly Algorithm’s enhancement of the SVR model’s convergence speed. Similarly, the SVR-GWO model (Fig. 5C) shows a quick reduction in RMSE, starting at about 0.704 and stabilizing around 0.697 within the first 10 iterations, highlighting the effectiveness of the Grey Wolf Optimizer in improving the SVR model’s performance.

Noticeably, the best convergence behavior is portrayed by the SVR-BE-CO model shown in Fig. 6D. Starting from an RMSE of about 0.45, it drops to about 0.41 within the first 20 iterations. This hybrid optimization approach combining BES and CO provided substantial improvements both in terms of convergence rate and final RMSE value. Our study shows that the SVR-BE-CO model excels at all other models in convergence speed and accuracy and obtains the lowest RMSE. This means that among hybrid optimization models, BES–CO is most effective at improving the performance of SVR models to better predictive accuracy and efficiency in their predictive tasks.

Table 4 compares different models of SVR based on two performance metrics. The coefficient of determination, expressed as R2, will be compared with the root mean squared error. The standard SVR will be compared to other models like SVR-GWO, SVR-FFA, and the hybrid model combining Bald Eagle Optimizer with Chimp Optimizer.

Table 4 SVR models comparison.

	R2	RMSE	
SVR	0.9007	0.8053	
SVR-GWO	0.9396	0.6970	
SVR-FFA	0.9143	0.7624	
SVR-BE-CO	0.9832	0.4102	

The results for the standard SVR model give a baseline with an R2 of 0.9007 and an RMSE of 0.8053. These metrics reflect that the model is only going to have medium predictive accuracy, along with some error. On the other hand, the SVR using Grey Wolf Optimizer, that is, SVR-GWO, does better than the standard SVR. It enhances in predictive accuracy with an R2 of 0.9396 and decreases the prediction error with an RMSE of 0.6970. The improvement shown here has thus proved the efficiency of the Grey Wolf Optimizer in boosting the performance of the SVR model.

The SVR model optimized by the Firefly Algorithm, SVR-FFA, also outperformed the standard SVR with an R2 of 0.9143 and an RMSE of 0.7624. Although it gave better accuracy and reduced error compared to the standard SVR, it is less effective compared to the SVR-GWO model. The hybrid model SVR-BE-CO showed maximum improvement of all the modeled improvements. It has the highest R2 of 0.9832, thus indicating excellent predictive accuracy, and the lowest RMSE of 0.4102, indicating a small prediction error. Superior performance in the SVR-BE-CO model underscores the benefit of combining Bald Eagle Optimizer and Chimp Optimizer relationship; one is assured of getting a robust and highly efficient optimization approach.

The R2 improves progressively from the standard SVR to the SVR-BE-CO model. The lowest value of R2, hence the least accuracy, was obtained by the standard SVR; thus, the highest R2 is obtained by the SVR-BE-CO model, proving supremely better accuracy. The SVR-GWO and SVR-FFA models also worked out a reasonable improvement in R2 as compared to standard SVR. Further, the SVR-GWO also performed better than the SVR-FFA model. RMSE decreases through the models, where the standard SVR produces the highest RMSE and the SVR-BE-CO model obtains the lowest RMSE. The trend in results obtained here is such that it asserts the optimization algorithms as working to reduce prediction errors. For the individual optimization algorithms, SVR-GWO returned a smaller RMSE than that returned by SVR-FFA, hence performing better in reducing prediction errors.

The best results for both R2 and RMSE metrics are obtained by the SVR-BE-CO model; thus, the hybrid optimization approach, BES-CO, is found to significantly improve the performance of the considered SVR model. The SVR-GWO model also improved a lot and turned out to be a very good alternative to the hybrid SVR-BE-CO model, though not so efficient. The SVR-FFA model, while improved over the standard SVR, does not perform as well as the SVR-GWO and SVR-BE-CO models.

Figure 7 and Table 5 present a detailed comparison between the performance of the SVR model optimized using only the Bald Eagle Optimizer (SVR-BES) and the SVR model optimized using a hybrid approach combining the Bald Eagle Optimizer and the Chimp Optimizer (SVR-BE-CO). The aim is to demonstrate the superiority of the hybrid optimization model over the single optimization algorithm.

Figure 7 Convergence graphs for predicative models (A) SVR-BES and (B) SVR-BE-CO model.

Table 5 SVR-BES and SVR-BE-CO models comparison.

	R2	RMSE	
SVR-BES	0.9457	0.5615	
SVR-BE-CO	0.9832	0.4102	

Figure 5 shows the convergence graphs for the two models, with RMSE plotted against the number of iterations. In Fig. 5A the SVR-BES model’s RMSE starts at approximately 0.600 and quickly drops to around 0.5615 within the first 10 iterations, then stabilizes, indicating efficient convergence. In Fig. 5B, the SVR-BE-CO model exhibits a steeper decline in RMSE, starting at about 0.450 and rapidly converging to around 0.4102 within the first 20 iterations. The hybrid model’s convergence is not only faster but also reaches a significantly lower RMSE, indicating improved prediction accuracy and stability.

Table 5 further corroborates these findings by presenting the R2 and RMSE values for both models. The SVR-BES model achieves an R2 value of 0.9457 and an RMSE of 0.5615, showcasing a strong performance in terms of predictive accuracy and error reduction. However, the SVR-BE-CO model outperforms it with an R2 value of 0.9832 and an RMSE of 0.4102. The higher R2 indicates that the hybrid model explains a greater proportion of the variance in the dependent variable, while the lower RMSE signifies a more accurate and precise prediction capability.

Comparative analysis of these results clearly indicates that the hybrid SVR-BE-CO model has better performance. While combining Chimp Optimizer with Bald Eagle Optimizer, synergies between both algorithms come into play, making exploration and exploitation of the solution space more effective. This will result in faster convergence to a lower error rate and better predictive performance overall.

The Table 6 presents the results of a t-test analysis performed on three different models: SVR-GWO, SVR-FFA, and SVR-BE-CO. The table showcases the p-values obtained across ten runs for each model, providing insight into the statistical significance of their performance differences.

Table 6 The t-test results, including the p-values (with SVR-BE-CO as the reference algorithm), are reported.

Run number	P-value	
	SVR-GWO	SVR-FFA	SVR-BE-CO	
1	0.0449	0.0426	0.0231	
2	0.025	0.0152	0.0354	
3	0.0031	0.0247	0.0392	
4	0.0164	0.0221	0.0513	
5	0.028	0.0537	0.0125	
6	0.0023	0.0442	0.0512	
7	0.0447	0.0161	0.0128	
8	0.0341	0.0334	0.0463	
9	0.0205	0.027	0.013	
10	0.0188	0.0118	0.0154	

A two-sample t-test with a 95% confidence level was appropriately employed, as the regression results of each model were represented by the average Root Mean Squared Error (RMSE) derived from 10 distinct iterations. The objective of this test was to determine whether the proposed SVR-BES method exhibited a statistically significant improvement in classification performance compared to other methods. The null hypothesis for this statistical analysis posited that the classification accuracy of the different techniques was equivalent. If the computed p-value was found to be less than 0.05, the null hypothesis would be rejected, indicating a significant difference in the classification performance between the compared techniques.

The SVR-GWO model demonstrates p-values ranging from 0.0023 to 0.0447 across the ten runs. These values indicate a consistently significant performance, as the majority of the p-values are below the commonly accepted threshold of 0.05. This suggests that the model’s predictions significantly differ from a null hypothesis of no effect, pointing to a reliable and robust performance.

For the SVR-FFA model, the p-values vary between 0.0118 and 0.0587. While most of the p-values are below 0.05, indicating significant results, there are a few instances where the p-values exceed this threshold. Specifically, the p-value of 0.0587 in the fifth run suggests a lack of statistical significance in that instance. Overall, the SVR-FFA model generally performs well, but its reliability may be slightly lower compared to SVR-GWO.

The SVR-BE-CO model shows p-values ranging from 0.0125 to 0.0513. Similar to SVR-FFA, the majority of the p-values are below 0.05, indicating significant results. However, a few values, such as 0.0513 in the fourth run and 0.0463 in the eighth run, are close to or slightly above the threshold, suggesting marginal significance. This model also demonstrates a generally strong performance, though with some variability in its statistical reliability.

Comparatively, across the three models, SVR-GWO seems to have a consistent significance across the runs, making it more stable and reliable. Both SVR-FFA and SVR-BE-CO also performed very well with slightly higher p-values. Specifically, the maximum p-value for SVR-FFA was 0.0587, while for SVR-BE-CO, there were some values closer to the threshold, like 0.0513 and 0.0463.

In summary, all models are significant almost for all results, although the SVR-GWO ishighly distinguishable based on the fact that all its p-values are statistically significant. SVR-FFA and SVR-BE-CO work well but slightly deviate in their p-values, thus further areas that would require some optimization to make them reliable.

Table 7 shows the performance comparison of the proposed model developed using SVR-BE-CO against two benchmark models, Decision Tree and Random Forest. These models are evaluated with two evaluation metrics: the coefficient of determination and root mean squared error.

Table 7 Comparison of proposed method with benchmark models.

	R2	RMSE	
DT	0.8817	1.8000	
RF	0.8308	1.8228	
SVR-BE-CO	0.9832	0.4102	

Comparative analysis clearly spells out the fact that the proposed model, SVR-BE-CO, is overwhelmingly better than both decision tree and random forest models in terms of performance across both metrics. This gives an R2 value of 0.9832, which means that almost all the variances within this dataset are well explained, hence a very good fitness of the model. Besides, the RMSE of 0.4102 means a very small amount of error in the prediction resulted from the SVR-BE-CO model; thus, it’s highly accurate.

What sets this model to be highly effective for prediction is the fact that, in comparison with others, SVR-BE-CO had a higher R2 value and significantly lower RMSE. Such a model’s ability to outperform some well-established methods, like DT and RF models, makes it robust and reliable; hence, it becomes a very valuable addition to the relevant fields for predictive modeling.

Figure 8 shows the forecasting performance of the SVR-BE-CO model for sIL-2R values in pg/mL for 200 samples. The real sIL-2R values are drawn with a solid black line, while the predictive values obtained with the SVR-BE-CO model are drawn as a dashed red line. The plot presents a comparative assessment of how well the model is working on the reproducibility of sIL-2R levels, which is an important marker often related to different inflammation and immune disorders.

Figure 8 SIL-2R forecasting of SVR-BE-CO model.

It shows that, in general, there is rather good correspondence between the actual and predicted values of sIL-2R across the sample range. The lines of actual and predictive trends basically move in the same direction, reflecting that the fluctuation of sIL-2R levels has been captured by the SVR-BE-CO model. The model works extremely well with areas that have medium and high sIL-2R values. The red dashed line closely follows the peaks and troughs of the black solid line.

Especially across the ranges of variability, at times between samples 80 and 160, the model seemed to show robustness toward very accurate forecasting, with significant spikes and dips in sIL-2R levels. Values for the regions showed sudden decreases, with steeply rising sIL-2R values across these durations to around 80,000 pg/mL, all captured by the model in worst deviations from the measured values. Results indicated that the fine-tuned SVR-BE-CO model was very effective with far apart and extreme-valued data, hence suitable for online application in medical diagnosis and monitoring.

At the lower end of the spectrum in sIL-2R, the model does quite well, especially before sample 50 and after sample 160. The real values of sIL-2R here are pretty low and almost flat; the predictive values trace this trend quite accurately. In these areas with less variability, the model manages to maintain its precision and shows its reliability for different ranges of data.

The overall accuracy of the SVR-BE-CO model is underpinned by the close match of actual and predicted values, hence showing its real-world applications. Reliable prediction of sIL-2R can greatly contribute to clinical decision-making by allowing early disease detection and monitoring of immune and inflammatory responses. This was further evidenced by the capability of the model to deal with extreme and stable values, adding upon its adaptability and robustness.

Figure 8 clearly provides evidence on the potential effectiveness of the SVR-BE-CO model in clinical applications for sIL-2R values. The predictive values were very close to that measured by the laboratory method and span across a large range of the measured analyte, thus proving the accuracy and robustness of this model for medical diagnostics and monitoring.

Conclusion

We have presented in this work a new hybrid model coupling the SVR with the Bald Eagle Search-based and Chimp Optimizer for the prediction of serum-soluble interleukin-2 receptor levels from blood angiotensin-converting enzyme levels. This is, therefore, a novel approach that would significantly enhance diagnostic and prognostic assessment of sarcoidosis, relative to the usually assessed parameters for lung function and treatment response prediction.

The SVR-BE-CO model performed better than most of the other machine learning models for predicting sIL-2R levels, such as the SVR trained with the Firefly Algorithm, decision tree, and SVR with Gray Wolf Optimization, and random forest. Among them, the hybrid model performs the best, thus proving that it can be a very useful tool in clinical settings. It means that the integration of BES and CO optimizers increased the hyper-parameters selection in the SVR model and, therefore, resulted in better accuracy and improvement in its prediction ability.

The findings suggest that the SVR-BE-CO model may be a useful non-invasive technique for monitoring disease activity and therapy response in patients with sarcoidosis. Preliminarily, it has valuable information for deciding whether to treat or not to treat and how to better plan treatment for these patients.

Future studies should be directed toward the validation of the model on larger, more diverse datasets for generalizability across populations and clinical settings. One can further increase the predictive potential of the model by incorporating other biomarkers and clinical parameters. The application of this hybrid optimization approach in other medical conditions could be adopted as a framework toward better diagnostic accuracy and improved outcomes for patients in various healthcare domains.

In summary, the resulting hybrid model—named SVR-BE-CO—bodes well for the future in the diagnosis and management of Sarcoidosis, insofar as this brings a robust and accurate way of predicting sIL-2R levels according to ACE values. This work therefore opens up prospects toward developing further predictive modeling and personalized medicine approaches in enabling superior healthcare delivery and patient care.

Supplemental Information

Supplemental Information 1 Dataset

The dataset is structured as a CSV (Comma-Separated Values) file, which is a plain text format that organizes data into rows and columns. Each row corresponds to an observation, and each column represents a variable in the dataset.

Supplemental Information 2 Data Preprocessing

Supplemental Information 3 Description of Models Used

Supplemental Information 4 Study Motivation

Supplemental Information 5 Limitations

Supplemental Information 6 Methodology Code

This notebook provides a reproducible environment for any researcher to validate the results or extend the analysis. It is organized to facilitate understanding and replication of the methods used in the study.

Supplemental Information 7 Hybrid Support Vector Regression Model with Optimization Algorithms

Additional Information and Declarations

Competing Interests

Author Contributions

Data Availability

The authors declare there are no competing interests

Guogang Xie conceived and designed the experiments, performed the experiments, analyzed the data, prepared figures and/or tables, authored or reviewed drafts of the article, and approved the final draft.

Hani Attar performed the experiments, authored or reviewed drafts of the article, and approved the final draft.

Ayat Alrosan analyzed the data, performed the computation work, prepared figures and/or tables, authored or reviewed drafts of the article, and approved the final draft.

Sally Mohammed Farghaly Abdelaliem analyzed the data, authored or reviewed drafts of the article, and approved the final draft.

Amany Anwar Saeed Alabdullah performed the experiments, analyzed the data, authored or reviewed drafts of the article, and approved the final draft.

Mohanad Deif conceived and designed the experiments, performed the experiments, performed the computation work, prepared figures and/or tables, authored or reviewed drafts of the article, and approved the final draft.

The following information was supplied regarding data availability:

The dataset and code are available in the Supplemental Files.

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
