# Peer review of "Enhanced diagnosing patients suspected of sarcoidosis using a hybrid support vector regression model with bald eagle and chimp optimizers"

_PeerJ Computer Science, doi:10.7717/peerj-cs.2455_

## Round 0.1 · original submission · Major Revisions

Dear authors,

Thank you for the submission. The reviewers' comments are now available. It is not suggested that your article be published in its current format. We do, however, advise you to revise the paper in light of the reviewers' comments and concerns before resubmitting it.

Best wishes,

·

Basic reporting

The explanation is unclear throughout the manuscript and needs professional English to be proofread.
The authors claim that this paper proposed a novel forecasting model for sIL-2R prediction, which combines Support Vector Regression (SVR) with the Bald Eagle Search (BES) algorithm. However, the following sections, Introduction, Related Work and Novelty and Contribution, are not convincing, and it is not clear what the motivation is. The author did not adequately describe Sarcoidosis to show this area's importance.
For example line 39: Suggested: Sarcoidosis is a disease characterized by the growth of tiny collections of inflammatory cells (granulomas) in any part of your body and most commonly the lungs and lymph nodes. But it can also affect the eyes, skin, heart and other organs. The cause of sarcoidosis is unknown, but experts think it results from the body's immune system responding to an unknown substance. There is no cure for sarcoidosis, but most people do very well with no treatment or only modest treatment. In some cases, sarcoidosis goes away on its own. However, sarcoidosis may last for years and may cause organ damage.
Section 1: The Introduction did not mention or discuss anything about the method used, particularly the SVM and SVR. Here, the focus is more on the domain, which is sarcoidosis.
Section 2: Related work is not well covered for both the method used and the domain.
Section 3: Novelty and contribution is not enough.
The structure follows to PeerJ standards.

Experimental design

The article content is within the Aims and Scope of the journal.
The investigation was not well performed and does not have high technical & ethical standards. The proposed method combines SVR and BSE and is applied to the soluble interleukin 2 receptor (sIL-2R). There is no modification on SVR or BSE; the authors implement the existing techniques to find the optimal value of the three parameters.

Since the authors reproduce the existing method, the code, dataset, computing infrastructure, and reproduction script is easily available from GitHub. But, yes, the authors showed how they utilized BSE in SVR via the flowchart. However, that is not adequate to consider as a novel method.

The authors did not discuss the dataset that they used in detail. No preprocessing was mentioned in the paper. The original file of the dataset provided by the authors showed that the variables or attributes do consist of variable attributes with different units. Here, preprocessing is very important.

The authors compared their proposed method with other optimization methods and described the evaluation and assessment metrics used.

Validity of the findings

The novelty of this work is not assessed as the authors apply the existing method of SVR and BES to sIL-2R. No new modification is found. The authors should try to modify or enhance the BSE to show some novelty. I believe there are still lops that can be further investigated.
This is the most important issue of the paper: novelty.

Additional comments

No

Reviewer 2 ·

Basic reporting

The authors present a hybrid forecasting model for measuring serum-soluble Interleukin-2 receptor (sIL-2R) in sarcoidosis patients; the authors’ model combines the optimization algorithm bald eagle search (BES) with SVR to improve the hyper parameters of the SVR model.

Experimental design

In section 4.1 is mentioned that the dataset employed was obtained from reference [44]. Provide more details about the dataset, for instance, number of instances, number of features, whether if the class instances is balanced, etc.

Validity of the findings

I recommend to separate the results and discussion section in two sections.

In the discussion section the authors should discuss the model proposed regarding other techniques. That is, if possible, perform experiments with the same dataset using the models mentioned in the related works section (neural networks, random forest, decision trees, etc.). The results obtained would make clearer the importance of the authors' contribution.

Additional comments

I suggest to present the bald eagle search as a pseudocode.

If possible, provide a link where the dataset can be downloaded.

---

## Round 0.2 · Minor Revisions

Dear authors,

Thank you for the revision. One of the original reviewers did not respond to the invitation for reviewing the revised paper. According to one final reviewer, your paper still needs a revision and we encourage you to address the minor concerns and criticisms of Reviewer 3 and resubmit your article once you have updated it accordingly.

Best wishes,

·

Basic reporting

I have no further comments.

Experimental design

No comment.

Validity of the findings

Acceptable.

Additional comments

I am satisfied with the corrected version of the paper.

Reviewer 3 ·

Basic reporting

See below

Experimental design

See below

Validity of the findings

See below

Additional comments

According to the revised paper, I have appreciated the deep revision of the contents and the present form of this manuscript. But there is still a little content, which need be revised according to the comment of reviewer in order to meet the requirements of publish. A number of concerns listed as follows:
1.Introduction seems to be incomplete. Please carefully check and supplement it.
2.The authors must clearly explain the difference(s) between the proposed method and similar works in the introduction. The authors should further highlight the manuscript's innovations and contributions.
3.The authors need to make a clear proofread to avoid grammatical mistakes and typo errors.
4.In order to highlight the introduction, some latest references should be added to the paper for improving the reviews part and the connection with the literature.

---

## Round 0.3 · accepted · Accept

Dear authors,

Thank you for the revised paper. The reviewers think that you have performed the necessary additions and modifications. Your paper now seems sufficiently improved and acceptable for publication.

Best wishes,

Reviewer 3 ·

Basic reporting

ok

Experimental design

ok

Validity of the findings

ok